# Effect of Gravity and Various Operating Conditions on Proton Exchange Membrane Water Electrolysis Cell Performance

**DOI:** 10.3390/membranes11110822

**Published:** 2021-10-27

**Authors:** Yena Choi, Woojung Lee, Youngseung Na

**Affiliations:** School of Mechanical and Information Engineering, University of Seoul, Seoul 02504, Korea; yenacfd@uos.ac.kr (Y.C.); dnwjd4548@uos.ac.kr (W.L.)

**Keywords:** two-phase flow, cell orientation, single serpentine, quintuple serpentine, bubble coverage, water electrolysis, operating temperature, flow rate, channel pattern

## Abstract

Water electrolysis is an eco-friendly method for the utilization of renewable energy sources which provide intermittent power supply. Proton exchange membrane water electrolysis (PEMWE) has a high efficiency in this regard. However, the two-phase flow of water and oxygen at the anode side causes performance degradation, and various operating conditions affect the performance of PEMWE. In this study, the effects of four control parameters (operating temperature, flow rate, cell orientation, and pattern of the channel) on the performance of PEMWE were investigated. The effects of the operating conditions on its performance were examined using a 25 cm^2^ single-cell. Evaluation tests were conducted using in situ methods such as polarization curves and electrochemical impedance spectroscopy. The results demonstrated that a high operating temperature and low flow rate reduce the activation and ohmic losses, and thereby enhance the performance of PEMWE. Additionally, the cell orientation affects the performance of PEMWE owing to the variation in the two-phase flow regime. It was observed that the slope of specific sections in the polarization curve rapidly increases at a specific cell voltage.

## 1. Introduction

Renewable energy conversion systems, such as photovoltaic systems and wind turbines, have received extensive attention. In the renewable energy generation process, the fluctuating supply of renewable energy sources produces nonuniform electrical power. The surplus power must be stored in an energy storage medium to avoid disturbing the power grid. The secondary batteries, such as lithium battery and flow battery, are typical energy storage media. Lithium batteries are expensive and flammable for large scale storage, and flow batteries have low energy density [1,2], while hydrogen, which can serve as a storage medium for surplus power, has high energy density (MWh, even TWh), and its storage duration is long [3]. Water electrolysis can produce green hydrogen using the surplus power in renewable energy generation [4,5]. Therefore, it is an environmentally friendly method to store surplus energy.

There are several types of water electrolysis, depending on the electrolyte. Among them, proton exchange membrane water electrolysis (PEMWE) is a suitable method for the utilization of renewable energy because its transient response occurs within approximately 50 ms. This characteristic is favorable when the input power is intermittent [6]. Additionally, it can operate at a high current density (above 1 A/cm^2^), which is highly effective [7]. There are four types of proton exchange membranes (PEMs): perfluorinated sulfonic acid (PFSA) and fluorinated, hydrocarbon (HC)-based, and functional PEMs. PEMs should satisfy low oxidant permeability and electrochemical thermal stability [8]. Among them, PFSA (Nafion) consists of polytetrafluoroethylene (PTFE, also known as DuPont’s Teflon™)-like backbone, side chains of which connect the molecular backbone to the third region, ion clusters consisting of sulfonic acid ions. Advantages of PFSA are stability for oxidative and reduction environments due to PTFE backbone structure and high protonic conductivities [9]. In this study, PFSA (Nafion) membrane was used.

In PEMWE, water is decomposed into protons, electrons, and oxygen by the oxygen evolution reaction at the anode, as expressed in Equation (1). Subsequently, the protons pass through the polymer electrolyte, and the electrons flow through an external circuit. Hydrogen is produced by the hydrogen evolution reaction at the cathode, as expressed in Equation (2). Oxygen produced at the anode side is discharged by the anode porous transport layer (PTL) and the channel. Therefore, a two-phase flow of the supplied water (liquid) and oxygen (gas) exists at the anode side. This two-phase flow regime affects the performance of PEMWE. The oxygen volume fraction increases with the increase in the current density. Oxygen disturbs the water diffusion in the anode channel and the PTL, leading to performance degradation [10].
(1)$$\mathrm{Anode:}\;H_{2}O\to 2H^{+}+2e^{-}+\frac{1}{2}O_{2}$$
(2)$$\mathrm{Cathode}:\;2H^{+}+2e^{-}\to \;H_{2}$$

The two-phase flow analysis can be conducted from two perspectives: microscopic and macroscopic. Microscopic analysis is related to the capillary behavior in the PTL [11], whereas macroscopic analysis is related to the channel [12].

Various studies have been conducted on the microscopic analysis of the two-phase flow. The two-phase flow in the PTL is significantly affected by the capillary action [13]. The capillary action and behavior of bubbles in the PTL have been calculated by one-dimensional mathematical modeling [14]. Panchenko et al. conducted a stoichiometric analysis of oxygen and water in a PTL by neutron radiography [15]. The microscopic behavior of two-phase flow has been studied, however, there is insufficient research on the macroscopic behavior of the two-phase flow.

Understanding the macroscopic behavior of the two-phase flow is crucial for the mass production of hydrogen. The active area should be expanded, and water should be distributed uniformly over the large active area to scale up PEMWE. The effect of mass transport should be considered, on which the two-phase flow in the channel has a much larger effect [16] than the PTL [17]. Therefore, it is crucial to evaluate the performance of PEMWE depending on the change in the two-phase flow regime and the operating conditions.

The operating conditions affect the two-phase flow regime and the performance of PEMWE. A previous study showed that the operating temperature and pressure of PEMWE affect the gas/liquid behavior, which affects its performance [18]. The two-phase flow in the channel and the performance vary with the operating temperature and the flow rate [19]. It is important to understand how the performance and the two-phase flow regime differ based on operating conditions.

In a polarization curve, an increment in the slope indicates an increase in the main overpotential sources. For enhancing the performance, it is important to analyze the change in the slope of the polarization curve. Majasan et al. studied the correlation between the micro-structure of a PTL and the performance of PEMWE [20]. The slope of a polarization curve initially increases and then gradually decreases. However, this phenomenon has not been extensively discussed. Schmidt et al. developed a two-phase model and analyzed the mass transport overpotential, which was compared to experimental results [21]. It was observed that the experimentally obtained overvoltage rapidly increased within a specific range. However, this range did not match with the model results, and the phenomenon could not be explained. Therefore, further studies should be conducted to investigate the rapid increase in the overpotential.

In this study, the performance of PEMWE was evaluated based on the effects of various operating conditions and the two-phase flow regime. Four parameters were selected, and experiments were performed by changing the variables. Initially, the effects of the operating temperature, flow rate, and pattern of the channel on the performance were studied. Additionally, the effects of gravity on the performance and the two-phase flow regime according to the cell orientation were investigated. Finally, the change in the slope of the polarization curve at a particular voltage was identified.

## 2. Materials and Methods

### 2.1. Materials

A PEMWE single-cell with an active area of 50 × 50 mm^2^ was designed and fabricated. The area of each bipolar plate was 80 × 80 mm^2^. Two types of anode channels were imprinted on the bipolar plates, and they were constructed from Pt-coated titanium. Figure 1 shows schematics of the channels: (a) a single serpentine, which has a single path from the inlet to the outlet, and (b) a quintuple serpentine, which has five paths with the same length. The cross-sectional area of both channels was 1 × 1 mm^2^. The pattern of the cathode channel was quintuple serpentine. The end plates were constructed using aluminum, and the current collectors using gold-coated copper. The dimensions of both end plates and current collectors were 120 × 120 mm^2^ and 80 × 80 mm^2^, respectively. The membrane-electrode assembly (MEA) consisted of N117 and two catalysts with an IrO_2_ (3.0 mg/cm^2^) anode and a PtB (1.0 mg/cm^2^) cathode (Eco C&T, Seoul, Korea). The PTL of the anode and cathode were composed of titanium papers: for anode, 2GDL06N-025 BS05PT, porosity 73%, and thickness 260 μm (BEKAERT, Zwevegem, Belgium), and for cathode, 2GDL09N-025 BS05PT, porosity 60%, and thickness 240 μm, (BEKAERT, Belgium) were employed. The gaskets of the anode and cathode were made of polytetrafluoroethylene glass fibers (Thickness 140 μm, CNL Energy, Seoul, Korea).

### 2.2. Experimental Procedure

A laboratory was constantly maintained at an indoor temperature of 26 °C and a humidity of 40% using a thermo-hygrostat. A single-cell was assembled by applying a torque of 15 N·m and using eight M10 bolts. The thickness of the PTLs (280 μm) was reduced by 56% compared to the previous thickness (500 μm). A commercial test station (water electrolysis system, CNL Energy, Korea) was utilized, and it consisted of heating and supplying reactants and exhaust products. The temperatures of the single-cell and those at the inlet and outlet of the channel were recorded. The experimental process is shown in Figure 2. The deionized water (18.22 MΩ·cm) in the reservoir was maintained at a temperature of 80 °C. Subsequently, water was injected into the anode channel inlet using a piston pump (SIMDOS FEM 1.10, KNF, Freiburg im Breisgau, Germany). A potential was applied using a potentiostat (HCP-803, BioLogic, Seyssinet-Pariset, France). Finally, the oxygen and hydrogen produced by the reaction were released as exhaust.

The temperature of the anode channel inlet was fixed at 55 °C for the experiments. The performance of the PEMWE single-cell was analyzed by conducting chronoamperometry (CA) and electrochemical impedance spectroscopy (EIS). To obtain the polarization curve, the voltage was increased stepwise (0.05 V) from 1.35 V to 2.40 V. The EIS measurements were conducted after obtaining the polarization curve, and they were performed at three voltages, 1.6 V, 2.0 V, and 2.4 V, in the frequency range from 10 kHz to 50 mHz. The amplitude was 14 mV. The impedance was recorded at an interval of 15 s or at a current exceeding 5 A. The polarization curve cycle was repeated five times to condition the single-cell. The performance did not change after the third cycle. Therefore, it was assumed to be sufficiently conditioned.

In this study, the performance of the PEMWE single-cell was analyzed based on the operating temperature, flow rate, cell orientation, and pattern of the channel. The values of the four parameters are listed in Table 1.

The first parameter is the operating temperature. It significantly affects the performance and leads to effects such as activation and ohmic overpotentials. The operating temperature is affected by the flow rate of the supplied water. Therefore, decoupling of the effects of the operating temperature on the activation and ohmic overpotential is required. Experiments were performed at operating temperatures of 75 °C, 80 °C, and 85 °C. The single-cell was heated using a heating plate, which was attached to the endplate. The temperature of the heating plate was regarded as the operating temperature. The second parameter is the flow rate of water. When the current density is changed, the ratio of water and oxygen and their velocities in the channel are also changed. This affects the heat transfer and PEMWE performance. The two-phase flow behavior in the channel of the PEMWE single-cell, having an active area of 9 cm^2^, was studied under various water flow rates (15 sccm, 30 sccm, 45 sccm, 60 sccm) [19]. The difference in the PEMWE’s performance between the water flow rate of 15 sccm and 45 sccm is wide. The flow rate (25 sccm, 50 sccm, 75 sccm) was selected considering the active area (25 cm^2^) and stoichiometric ratios of two and three. Regarding the third parameter, six orientations of the PEMWE cell were considered. The orientation was defined according to the direction of the anode channel. Figure 3 depicts the various cell orientations, where “+” indicates that the inlet is located above the outlet, and “−“ denotes the opposite. “X” signifies that the direction of the flow path is horizontal with respect to the ground, and “Y” represents that it is vertical. “Z” indicates that the MEA plane and the ground are horizontal. The final control parameter is the pattern of the channel. To investigate its effects on the performance, experiments were conducted using single and quintuple serpentine channels.

## 3. Results and Discussion

### 3.1. Operating Temperature

Experiments were conducted at operating temperatures of 75 °C, 80 °C, and 85 °C. An electrolysis cell with a single serpentine anode channel pattern was oriented in the X+ direction, and water was supplied to the anode channel at a flow rate of 25 sccm. Figure 4a shows the polarization curves obtained with each operating temperature. At the same voltage, when the operating temperature was increased, the current density increased. At a cell voltage of 2 V, the current density of the electrolysis cell at an operating temperature of 85 °C (1.887 A/cm^2^) was 11.1% higher than of that at 75 °C (1.660 A/cm^2^). Each value of resistance was measured in the Nyquist impedance spectrum at 1.6 V, as shown in Figure 4b. The area-specific impedance decreased with the increase in the operating temperature. The ohmic resistance at 85 °C (0.210 Ω·cm^2^) was 6.7% lower than that at 75 °C (0.224 Ω·cm^2^). The area-specific impedance is affected by ion conductivity and membrane thickness. Because the same MEA was used in each experiment, the change in the ion conductivity resulted in the above change in the area-specific impedance. The ion conductivity is calculated using Equations (3) and (4) [22].
(3)$$\sigma \left(T,\lambda \right)=\sigma_{303K}\left(\lambda \right)\exp\left[1268\left(\frac{1}{303}-\frac{1}{T}\right)\right]$$
(4)$$\sigma_{303K}\left(\lambda \right)=0.005193\lambda -0.00326$$
where σ is the ion conductivity, T is the operating temperature, and λ is the water content of the membrane.

In PEMWE, the Nafion membrane is assumed to be completely hydrated. Therefore, the water content was assumed to be 22. The ion conductivities were 0.191 S/cm, 0.2 S/cm, and 0.211 S/cm at the operating temperatures of 75 °C, 80 °C, and 85 °C, respectively. When the operating temperature was raised by 10 °C, the ion conductivity increased by 10.5%. Therefore, a high operating temperature enhances the ion conductivity and increases the performance.

Additionally, the diameter of the arc decreases with the increase in the operating temperature, as shown in Figure 4b. This indicates that the activation overpotential decreased with the increase in the operating temperature. The exchange current density affects the activation overpotential, and it increases exponentially with the increase in the operating temperature. Increment in the exchange current density improves the reaction rate, thereby enhancing the performance.

### 3.2. Water Flow Rate

The PEMWE performance was evaluated at flow rates of 25 sccm, 50 sccm, and 75 sccm and at an operating temperature of 80 °C. The single serpentine anode channel was oriented in the X+ direction. The polarization curves for each flow rate are shown in Figure 5a. At the same voltage, the current density increased with the decrease in the flow rate. At 2 V, the current density of PEMWE at 25 sccm was 14.8% higher than that at 75 sccm. Figure 5b shows the Nyquist impedance spectrum at 1.6 V. The area-specific impedance increased with the increase in the flow rate. The cooling rate depending on the flow rate affects the area-specific impedance. Because the anode channel inlet temperature was fixed at 55 °C and the heating plate temperature was 80 °C, the single-cell was cooled by supplied water. Therefore, the cooling rate increased with the increase in the flow rate. Hence, the operating temperature of the single-cell decreased. It can be observed from Figure 6 that when the flow rate is high, the anode outlet temperature is low. This phenomenon reduces the ion conductivity and thereby increases the area-specific impedance. Additionally, the arc diameter increases when the flow rate is increased. The activation overpotential might be increased with the decrease in the operating temperature. Activation overpotentials of 0.325 V, 0.329 V, and 0.33 V, as calculated by Tafel fitting [22], were obtained at the flow rate of 25 sccm, 50 sccm, and 75 sccm, respectively.

### 3.3. Cell Orientation 

A two-phase flow of water (liquid) and oxygen (gas) occurs in the anode channel in PEMWE. The density of water in the liquid state is 890 times higher than that of oxygen in the gaseous state at 80 °C and 1 atm. Oxygen locates above the water surface, owing to buoyancy. If the orientation of the PEMWE cell is changed, the positions of water and oxygen in the channel will also change, which can affect the PEMWE’s performance. Experiments with varied cell orientations were conducted, with three different water flow rates using the single serpentine anode channel at an operating temperature of 80 °C. The polarization curves of the different cell orientations are shown in Figure 7. A difference in the performance was observed at a cell voltage of 2 V. At 2 V, the highest PEMWE performance was in the Z direction. The PEMWE’s performance presented the descending order of X− > Y+ > Y− > X+. The maximum performance difference between Z+ and X+ was 23.17% at a flowrate of 75 sccm. 

The two-phase flow regime can affect the performance of an electrolysis cell. At a low current density, the quantity of water is dominant in the channel. Consequently, the flow regime in the channel is stationary, even if the cell orientation is changed. However, as the current density increases, oxygen starts accumulating in the channel [19]. When the cell orientation is changed, the positions of water and oxygen in the channel vary depending on the direction of the buoyant force. This affects the reactant concentrations at the reaction sites and increases the mass transport resistance. In this study, the mass transport resistance obtained by EIS at 1.6 V and 2.0 V in the case of 25 sccm flow rate are shown in Figure 8. At 1.6 V, a slight difference was noticed at different cell orientations. However, at 2.0 V, the impedance tails differed with the direction. It was hypothesized that the electrochemical surface area (ECSA) changed based on the cell orientation. The difference in the two-phase flow regime can affect the ECSA and the PEMWE performance. At the high current density (over 2 A/${\mathrm{cm}}^{2}$), the volume fraction of oxygen in the channel and PTL is increased. Even if the cell orientation changes, the two-phase flow regime is similar because oxygen gas dominates the flow channels. The performance of PEMWE is identical at the high current density.

When the electrolysis cell was oriented in the Z+ direction, water flowed to the PTL under the effect of gravity and oxygen flowed to the channel under the buoyancy effect. In the case of Z− direction, although the PTL was located above the channel, it contained a relatively large volume of water, and the channel consisted of a large volume of oxygen under the capillary force effect. Therefore, water and oxygen were uniformly distributed over the channel in the Z direction compared with the distribution in the other cell orientations. This phenomenon is depicted in Figure 9c,f. An increase in the humid area of MEA reduced the ohmic overpotential and improved the performance. Concurrently, the X and Y directions, flow field plate, and gravity force were non-orthogonal. Oxygen should flow along the buoyancy direction in the channel and the PTL. Therefore, oxygen accumulated in the local area instead of in the active area. Particularly, oxygen was located in the upper portion of the channel and water was in the lower portion of the channel, which is shown in Figure 9a,b,d,e.

In the X− direction case, the outlet was located on the upper side, and the buoyancy and flow directions were coincident. Conversely, the X+, buoyancy, and flow directions were opposite. Therefore, the water distribution was blocked by oxygen at the inlet. Oxygen could be more easily discharged in the X− direction than in the X+ direction. Therefore, the active area in the X− direction can be larger than that in the X+ direction, and the performance in the X− direction is higher than that in the X+ direction, as shown in Figure 9a,d.

In the Y+ direction case, although the upper section of the active area was mostly filled with oxygen, the overall active area was wet because water was supplied from the top of the channel, which flowed following the direction indicated by the arrow in Figure 9b. Conversely, in the Y− direction, because water was supplied from the bottom of the channel, the active area was divided into two parts: upper part filled with oxygen and lower part filled with water. Therefore, the volume fraction of water in the PTL in the Y+ direction was greater than that in the Y− direction, and the humid area in the Y+ direction was larger than that in the Y− direction. The PEMWE’s performance was better in the Y+ direction than that in the Y− direction. 

### 3.4. Anode Channel Pattern

The cell orientation affects the PEMWE’s performance with the single serpentine channel for the anode, as described in Section 3.3. However, the PEMWE’s performance with the quintuple serpentine channel is only slightly affected by the cell orientation, as shown in Figure 10. The polarization curves were almost similar. The difference in the performance based on the cell direction was in the range of 0.5–2.7%, which was negligible. 

The two-phase flow regime in the channel can be estimated by calculating the velocity of each fluid. The velocity of water at the anode inlet was calculated, and the velocity of oxygen at the outlet was calculated using Faraday’s law at 2 A/${\mathrm{cm}}^{2}$. Subsequently, the two-phase flow pattern for each channel was estimated using a two-phase flow regime map in horizontal pipes. [23] The two-phase flow regime of the quintuple serpentine might be between a stratified flow and a wavy flow. When the pressure drops of the five paths differ, the water flows through the low-pressure drop paths. Therefore, the five paths can be divided into water flow paths and oxygen flow paths. If the cell orientation is changed, the low pressure drop paths are also shifted. Irrespective of the cell orientations, a similar area of the channel is used for the electrochemical reaction. In the case of the quintuple serpentine channel, a slight difference in the PEMWE performance was observed based on the cell orientation.

The average PEMWE performance for the experiments with the single serpentine channel (1.448 A/${\mathrm{cm}}^{2}$) was 8.3% higher than that with the quintuple serpentine channel (1.336 A/${\mathrm{cm}}^{2}$). The performance varied owing to the geometry characteristics of the channels. The pressure drop of the single serpentine channel was higher than that of the quintuple serpentine channel. Water flowed under the ribs to expand the active area. In the quintuple serpentine channel with five paths, the flow rate and the length per a single path were reduced. The pressure drop was lower than that in the single serpentine channel. A small volume of water diffused into the catalyst layers, which reduced the active area, and the performance was lower than that of the single serpentine cell.

The slope of the polarization curves of the PEMWE cell with the single serpentine channel is increased significantly at certain voltages, as shown in Figure 7. It can be observed from Figure 11 that the slope for the PEMWE cell with the single serpentine channel is significantly increased, whereas that for the quintuple serpentine channel is slightly increased. The slope was calculated from the ratio of the cell voltage change to the current density change between two consecutive measurement points on the polarization curve. The slope values were maximum at the peaks in both channels, where the cell voltage was approximately similar in all cases. This indicates that maximum overpotential changed at a specific voltage.

The activation, ohmic, and mass transport overpotential were determined from the experimental results of the current density and EIS measurements. The activation overpotential was calculated by Tafel fitting [22], and the ohmic overpotential was determined using EIS and the current density data. The remaining overpotential was regarded as the mass transport overpotential. Figure 12 shows the activation and ohmic overpotentials with the single and quintuple serpentine channels. The activation overpotentials of both channels was approximately similar at the operating conditions; this is shown in Figure 12a,b. The ohmic overpotentials varied; this is displayed in Figure 12c,d. These trends indicate that the activation overpotential was causative of the maximum slope, rather than another overpotential. When the activation overpotential reached a specific value, the overpotential was rapidly increased. The activation overpotential is associated with ECSA [24], which, in turn, is related to the oxygen coverage on the catalyst layer. In these experiments, if oxygen covered the catalyst layer, which reduced the ECSA, the overpotential was significantly increased. The slope change was maximum in the polarization curve. 

An identical case was observed for the quintuple serpentine channel. A maximum slope at a constant voltage was observed. The voltage at the maximum slope was in the range of 2.15–2.3 V in the case of the quintuple serpentine channel. However, it was in the range of 1.9–2.1 V for the single serpentine channel. At these points, the activation overpotential at the maximum slope was 0.37 V–0.38 V for the quintuple serpentine channel. This is greater than that of the activation overpotential of the single serpentine channel (0.31–0.33 V). These results indicated that the ECSA of the PEMWE cell with the quintuple serpentine channel is smaller than that of the single serpentine channel. A relatively large area of the catalyst layer in the PEMWE cell with the quintuple serpentine channel was covered with oxygen, compared with that of the single serpentine channel. This was observed because the intrusion under the rib was weaker for the quintuple serpentine channel than that for the single serpentine channel. This might explain the reduced ECSA of the quintuple serpentine channel.

## 4. Conclusions

The effects of various parameters on the performance of a PEMWE cell were experimentally analyzed to enhance the cell performance. The effects of the operating temperature, water flow rate, pattern of the anode channel, and cell orientation were investigated. 

When the operating temperature was increased, the performance of PEMWE was improved. The activation overpotential was reduced owing to the increase in the exchange current density. This enhanced the electrochemical reaction. The ohmic overpotential was reduced because the ion conductivity of the membrane increased with the increase in the temperature.

Because the inlet temperature was lower than the cell temperature, a higher flow rate of water decreased the cell temperature. The cooling rate between the water and the cell increased when the flow rate was increased. This led to an increase in the area-specific impedance, which reduced the performance of the cell.

The cell orientation affected the performance at a high current density for the single serpentine channel because the two-phase flow regime of water and oxygen in the channel and the PTL varied with the cell orientation. However, this did not affect the performance of the PEMWE cell with the quintuple serpentine channel because the active areas did not vary with the cell orientation.

The slope of the polarization curve had a peak at a specific voltage. The maximum slope on the polarization curve was observed at approximately 2.0 V for the single serpentine channel and 2.2 V for the quintuple serpentine channel for all operating conditions. This phenomenon was due to the bubble coverage on the catalyst layer; however, further studies are required to clearly explain it.

It is crucial to determine the optimal operating conditions of PEMWE to enhance its performance. It is suggested that the bubble coverage on the catalyst should be reduced during the operation of the PEMWE cell. If the PEMWE cell operates at a low current density, a single serpentine anode channel is effective. Additionally, a quintuple serpentine channel is appropriate for operating the PEMWE cell at a high current density, because the ECSA is rapidly reduced at a relatively high cell voltage relative to that in the single serpentine channel case.

## Figures and Tables

**Figure 1 membranes-11-00822-f001:**
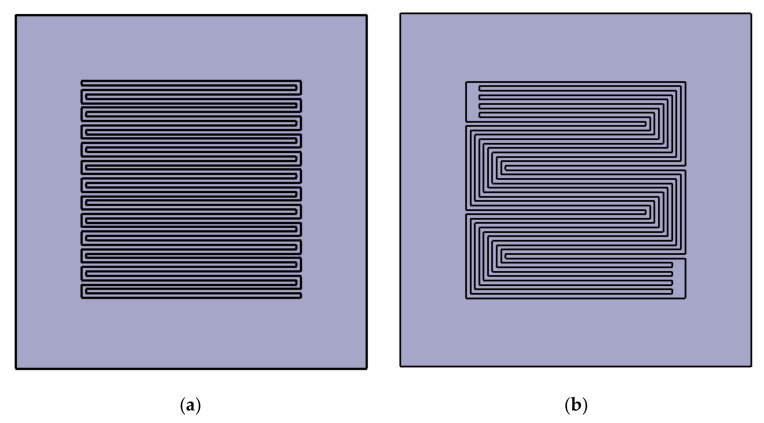
Schematic of the anode channel patterns (**a**) with the single serpentine channel and (**b**) with the quintuple serpentine channel machined on Pt-coated titanium bipolar plates.

**Figure 2 membranes-11-00822-f002:**
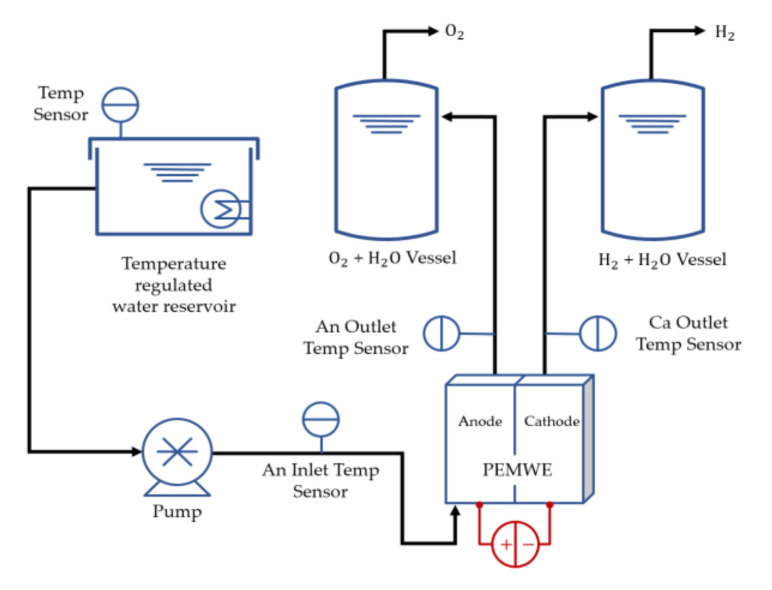
Schematic of the experimental setup of PEMWE using a pump, the reservoir, and temperature sensors.

**Figure 3 membranes-11-00822-f003:**
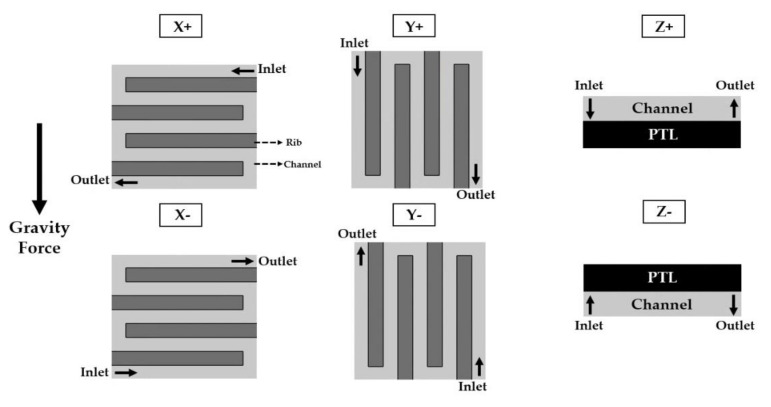
Schematic of PEMWE cell orientation according to the anode channel direction.

**Figure 4 membranes-11-00822-f004:**
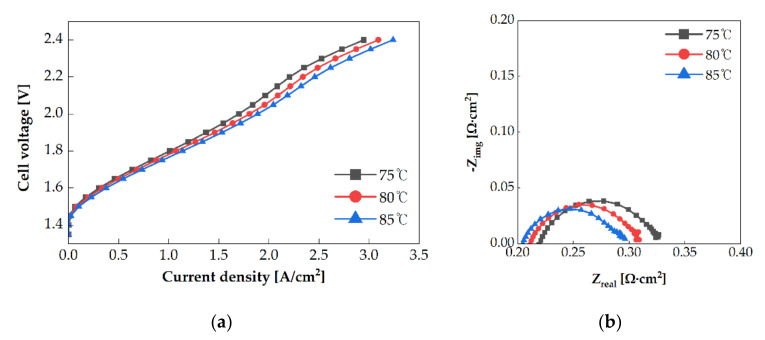
(**a**) Polarization curves of PEMWE and (**b**) electrochemical impedance of PEMWE at 1.6 V using the single serpentine anode channel at the different operating temperatures.

**Figure 5 membranes-11-00822-f005:**
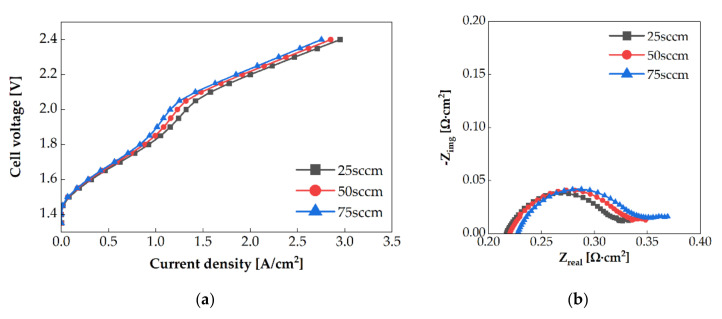
(**a**) Polarization curves of PEMWE and (**b**) electrochemical impedance of PEMWE at 1.6 V using single serpentine anode channel at the different flow rates.

**Figure 6 membranes-11-00822-f006:**
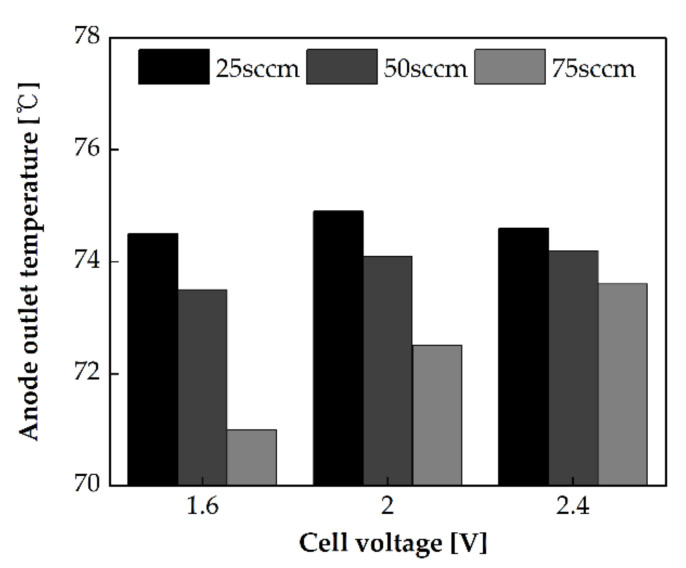
The anode outlet temperature at a voltage of 1.6 V, 2.0 V, and 2.4 V, and different flow rates in the single serpentine anode channel oriented in the X+ direction.

**Figure 7 membranes-11-00822-f007:**
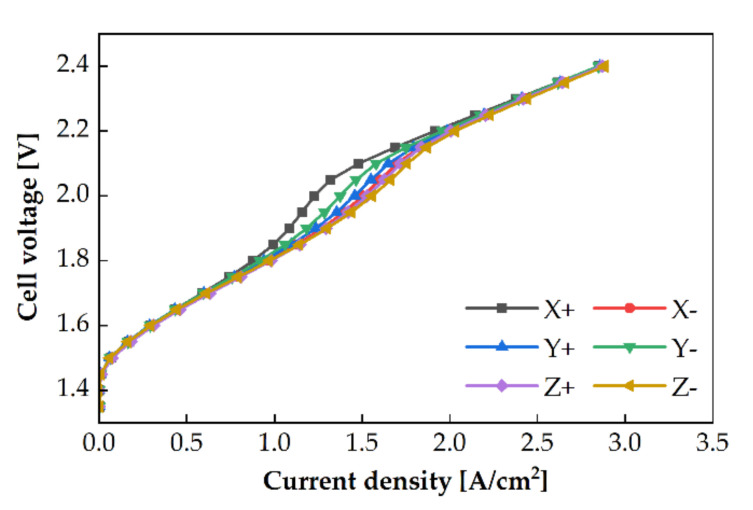
Polarization curves of PEMWE for different cell orientations at the water flow rate of 25 sccm to the single serpentine anode channel.

**Figure 8 membranes-11-00822-f008:**
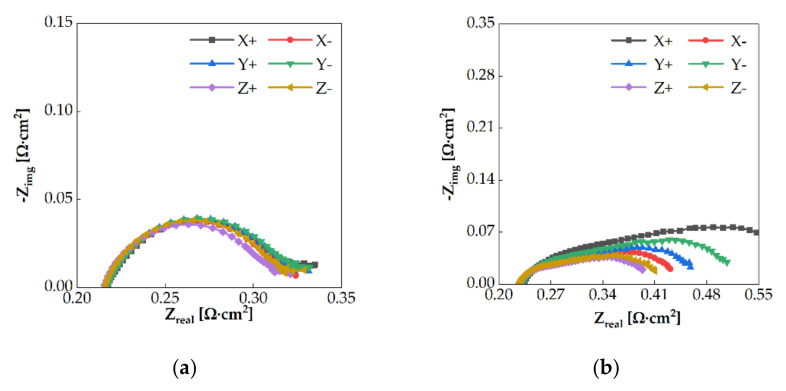
EIS for each cell orientation for the water flow rate of 25 sccm to the anode single serpentine channel at cell voltage of (**a**) 1.6 V and (**b**) 2.0 V.

**Figure 9 membranes-11-00822-f009:**
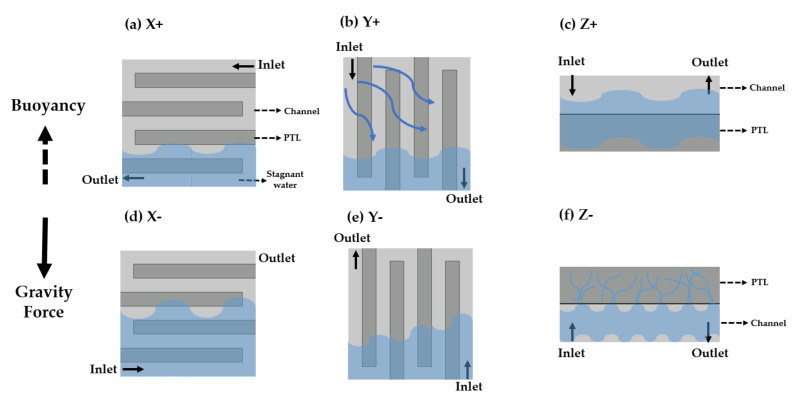
Schematic of two-phase flow regime in the channel and PTL of a PEMWE single-cell based on the cell orientation.

**Figure 10 membranes-11-00822-f010:**
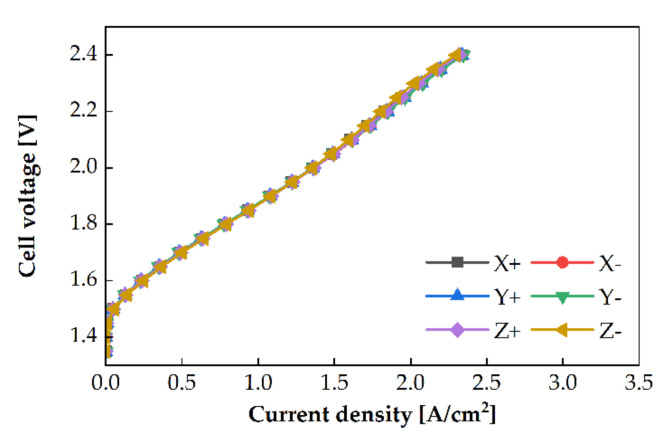
Polarization curves of PEMWE with the quintuple serpentine anode channel for the cell orientations at the water flow rate of 25 sccm and the operating temperature of 80 °C.

**Figure 11 membranes-11-00822-f011:**
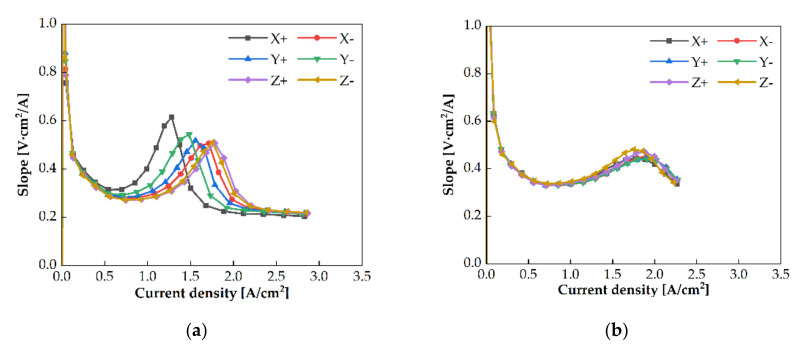
The slopes of polarization curves based on the current density with (**a**) the single serpentine channel and (**b**) the quintuple serpentine channel as the PEMWE anode channel.

**Figure 12 membranes-11-00822-f012:**
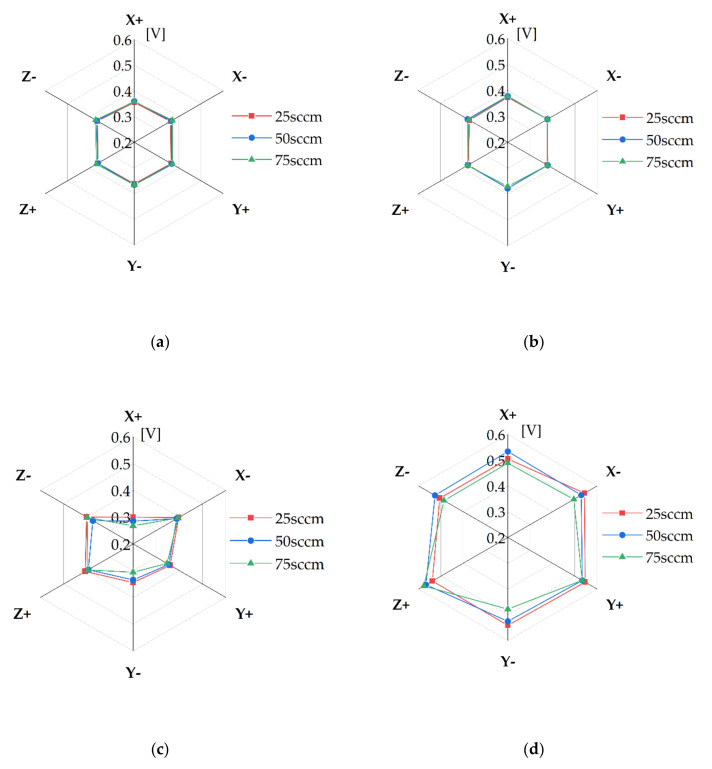
The activation overpotential with (**a**) the single serpentine channel and (**b**) the quintuple serpentine channel and the ohmic overpotential with (**c**) the single serpentine channel and (**d**) the quintuple serpentine channel for the experiments.

**Table 1 membranes-11-00822-t001:** Control parameters for operating PEMWE under various operating conditions.

Parameter	Value
Operating temperature	75 °C, 80 °C, 85 °C
Flow rate	25 sccm, 50 sccm, 75 sccm
Cell orientation	X+, X−, Y+, Y−, Z+, Z−
Pattern of channel	Quintuple serpentine, Single serpentine

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
