# Peer review of "Effect of Gravity and Various Operating Conditions on Proton Exchange Membrane Water Electrolysis Cell Performance"

_membranes, 2021, doi:10.3390/membranes11110822_

Round 1
Reviewer 1 Report
The manuscript reported the effect of various parameters on the performance of a PEMWE cell in order to enhance the cell performance. The various parameters include operating temperature, water flow rate, the pattern of the flow channel, and cell orientation. This work especially focuses on the issue of the two-phase flow of water and oxygen at the anode side, which leads to performance degradation of PEMWE. In-situ methods such as polarization curves and electrochemical impedance spectroscopy were tested for evaluation. A high operating temperature and a low flow rate may reduce the activation and ohmic loss, and enhance the performance. The cell orientation affects the performance due to the variation in the two-phase flow regime.
I consider the content of this manuscript will definitely meet the reading interests of the readers of the Membranes journal. However, there are certain English spelling and grammar issues, and also the discussion and explanation should be further improved.
Therefore, I suggest giving a minor revision and the authors need to clarify some issues or supply some more experimental data to enrich the content. This could be a comprehensive work after revision.
My detailed comments can be found in a separate PDF file.

Author Response
We could have a valuable chance to correct our manuscript. The authors would like to deeply thank the reviewers for that. We have revised it according to the reviewers’ comments. Enclosed please find the revised manuscript. The following are the point to point answers for the comments from the reviewers.

Reviewer 2 Report
The report titled "The effect of gravity and various operating conditions on the performance of proton exchange membrane water electrolysis cells" describes very insightful studies about the effect of two-phase flow (water and oxygen) with various operating conditions (operating temperature, flow rate, cell orientation, pattern of channel) on the performance of PEMWE. The method and the subject matter are very interesting to the journal and the broad spectrum of its readers. As a reviewer, I would like to refer authors to consider the following suggestions to improve the manuscript quality.
- In section 3.3, the authors hypothesized that the electrochemical surface area (ECSA) changes according to the cell orientation. It seems like an important parameter to determine the behavior of PEMWE performance depending on orientation. For a wide range of readability, authors should include those values.
- In figure 7, it was shown that the cell orientation does not affect the cell performance at high current density and voltage regions. Why? The authors have not explained the behavior.
Author Response

(The authors gave the same response as above.)
